# Global and Regional Impacts of Forest Expansion on Future Wildfires

James A. King¹\*, Kimberley J. Simpson², Christopher G. Bousfield², and Maria Val Martin¹\*

5

- 1 Leverhulme Centre for Climate Change Mitigation, School of Biosciences, University of Sheffield, Sheffield, UK
- 2 Ecology and Evolutionary Biology, School of Biosciences, University of Sheffield, Sheffield, UK

\* Authors to whom correspondence should be addressed – <u>james.king@sheffield.ac.uk</u>, m.valmartin@sheffield.ac.uk

#### 35 Abstract

Expansion of global forest cover via afforestation, reforestation, and forest restoration is a widely proposed nature-based solution for climate change mitigation, yet its effect on wildfire activity is poorly understood. As anthropogenic climate change intensifies and wildfire regimes change globally, evaluating the interactions between forest expansion, climate and population change is critical. We assess how large-scale forest expansion influences future fire activity and the land carbon sink using the Community Land Model version 5 (CLM5) with a mechanistic fire module. We simulate a maximum forest scenario (~750 Mha by 2100) under 2°C and 4°C warming pathways and compare it to three different land use trajectories with varying levels of forest cover and population change. We find that tropical forest expansion decreases fire activity by halting deforestation fires and replacing flammable grasslands with less flammable tree cover. In contrast, temperate forest expansion, such as in the Mediterranean, central Asia and continental US, can more than double fire carbon emissions under high warming, due to drier conditions and increased fuel loads. Population changes also influence fire regimes, with rising population growth in sub-Saharan Africa suppressing fire and reducing burned area, while decreasing populations in Europe and parts of East Asia are associated with increased fire activity. Finally, fires reduce the global land carbon sink by up to ~60 PgC by 2100, equivalent to ~5.6 times presentday annual CO2 emissions, emphasising the need to incorporate fire into climate mitigation planning. Our results suggest that forest expansion can both reduce and intensify fire risk depending on location and that fire-climate-land-human feedbacks must be accounted for in nature-based CO<sub>2</sub> mitigation strategies.

#### 60 1. Introduction

Fire is an important component of the Earth system. It is one of the most significant controls on the carbon cycle, affecting terrestrial and atmospheric carbon storage, and undergoing complex feedback responses with weather, climate, water, and vegetation (Li et al., 2018). Fire is also of great importance for human society; fire impacts include health implications of poor air quality (Val Martin et al., 2015; Silver et al., 2024; Tang et al., 2025; Shi et al., 2025), loss and damage to buildings and infrastructure (Kim et al., 2023), and exacerbation of anthropogenic climate change due to release of greenhouse gases and loss of carbon sinks (Nolan et al., 2021; Allen et al., 2024; Park et al., 2023; Liang et al., 2025). Understanding the drivers and impacts of fire is therefore crucial both environmentally and socioeconomically (Haas et al., 2022).

80

90

105

110

Global burned area has declined in recent decades (Andela et al., 2017), driven largely by anthropogenic fire suppression as a result of population growth, economic development, and agricultural expansion (Li et al., 2018). However, there has not been a corresponding decrease in fire emissions (Li et al., 2018) due to an increasing signal in emissions driven by high-biomass forest fires. Forest ecosystems, with greater carbon density, produce more carbon intensive emissions per unit area burned than grassland (Zheng et al., 2021). Consequently, increasing forest fires, driven by anthropogenic climate change on regional (Turco et al., 2023; Abatzoglou et al., 2025) and global (Abatzoglou et al., 2019) scales, are offsetting the decrease in grassland fires which dominate global burned area. Recent estimates suggest a 60% increase in forest fire emissions over the period 2001-2023, primarily in extratropical regions (Jones et al., 2024).

These trends raise questions about the long-term effectiveness of forest-based climate mitigation (Romm et al., 2025). Large-scale forest expansion, via afforestation, reforestation and/or restoration, has been widely proposed as a nature-based solution in the context of carbon dioxide removal (CDR) (Girardin et al., 2021; Lewis et al., 2019; Griscom et al., 2017). Forests are central to national and international climate strategies, including contributions to the Paris Agreement, where ~25% of planned CO<sub>2</sub> emission reductions rely on forest carbon sequestration (Grassi et al., 2017; Roe et al., 2021). International commitments to restore forest cover, such as the Bonn Challenge and the UN Decade on Ecosystem Restoration, have explicitly framed forests as a CDR strategy, and have been joined in recent years by a number of national-level, NGO-led, and private sector forest expansion initiatives (Seddon et al., 2021).

At the same time, there is growing concern that forestation could produce unintended environmental consequences. These include albedo and atmospheric chemistry feedbacks (Weber et al., 2024; Allen et al., 2024), increasing water demand (King et al., 2024; Hoek van Dijke et al., 2022; Bentley and Coomes, 2020), and shifts in atmospheric circulations (Portmann et al., 2022; De Hertog et al., 2024; Laguë and
 Swann, 2016; Swann et al., 2012). Fire adds an additional layer of complexity and must be considered when evaluating the suitability of forestation as a CDR strategy both globally and locally.

Reforestation increases ecosystem carbon density, potentially leading to increased fire emissions in fire-prone regions (Choi et al., 2006). This is of particular concern in the extratropical temperate and boreal forests, where wildfires have been increasing due to climate change (Jones et al., 2024), and are projected to increase under further warming (Cunningham et al., 2025). Tropical rainforests, such as the Amazon, are projected to experience increases in extreme fire weather under climate change (Abatzoglou et al., 2025), reducing resilience to climate variability and change, and potentially contributing towards future large-scale collapse followed by conversion to grassland (Flores et al., 2024). Conversely, the encroachment of trees and shrubs into savanna grasslands (Stevens et al., 2016), which rely on fire to maintain ecosystem structure and function, may reduce fire frequency (Venter et al., 2018) but increase fire severity (Stevens and Bond, 2023). Forest cover change also interacts with population

density; increases in population incentivise fire suppression (Knorr et al., 2014) whereas depopulation of rural areas can reduce fire suppression (de Diego et al., 2023).

Despite proposals to expand forest cover, the influence of fire as a counteracting driver remains poorly addressed in global climate mitigation assessment. The potential for forest expansion to have different effects on fire regimes across regions-underscores the need for mechanistic fire models to investigate possible future trajectories (Hantson et al., 2020).

To address these gaps, we use a fire model to investigate how fire may be affected by a plausible global forest expansion scenario under different warming levels. We tackle four key objectives: (1) address the impact of forest expansion on burned area and fire emissions on a global scale, (2) examine this impact in more detail on a regional scale, (3) assess the importance of human population change on fire activity and (4) quantify the effect of fire on the future carbon sequestration potential of the land carbon sink.

#### 2. Methods

120

125

#### 2.1 Model description

We used version 5 of the Community Land Model (CLM5; Lawrence et al., 2019) at a global 0.9° x 1.25° horizontal resolution. The model was run in 'BGC-Crop' mode with prognostic vegetation and active biogeochemical cycling. CLM5 included an active fire module (Li et al 2012, Li and Lawrence, 2017), which has participated in the Fire Model Intercomparison Project (FireMIP; Li et al., 2019).

The fire module simulates burned area as a function of weather and climate conditions (e.g. lightning frequency and wind), vegetation conditions (e.g. root-zone soil moisture and plant functional type (PFT) properties), and anthropogenic ignition and suppression. Human influence is parameterised as functions of population density and gross domestic product. Following the burned area calculation, the fire module calculates changes in land C and N pools due to fire, including speciated fire C emissions, and updates these pools in land ecosystems in the wider model. Four fire types are simulated: deforestation fires (tropical closed forests), peatland fires, agricultural fires (croplands), and non-peat fires outside cropland and tropical closed forests. This module is used in a current state-of-the-art Earth System Model, CESM2 (Li and Lawrence, 2017; Danabasoglu et al., 2020), and in recent studies of global fire dynamics under climate change (Tang et al., 2023; Liang et al., 2025; Shi et al., 2025; Bhattarai et al., 2025).

#### 2.2 Model experiments

The modelling procedure followed standard experimental protocols for CLM5 (Figure 1; NCAR, 2020). First, an initial spin-up was undertaken from arbitrary initial conditions

in which the atmospheric forcing cycled over pre-industrial conditions for 400 model years in 'accelerated decomposition' mode until carbon pools reached equilibrium. Next, the 'final spinup' stage was run (with all model components active) for 2000 model years until the carbon pools had again reached equilibrium criteria, with <3% of the land surface in disequilibrium. Following this, a transient simulation was run from 1850-2014, using historical land use/land cover (LULC) data compiled for the Coupled Model Intercomparison Project phase 6 (CMIP6; Lawrence et al., 2016) and atmospheric forcings from the Global Soil Wetness Project phase 3 (GSWP3; Kim, 2017).

To provide a 'present day' starting point for experimental model runs, we then ran the model from 2015-2025 following SSP2-4.5 LULC and climate. This represents a continuity scenario in which global trends do not change significantly from recent historical patterns (Riahi et al., 2017). We then initialised our experimental runs from the end point of this SSP2-4.5 run.

We applied atmospheric forcings, including wind, air temperature, precipitation, and CO<sub>2</sub> concentrations, following an 'anomaly forcing' approach. Forcings were derived from historical GSWP3 data covering the period 2001-2013, to which anomalies derived from fully-coupled runs of CESM2 following SSP-RCP scenarios were used. This approach, which is standard procedure for CLM (Lawrence et al., 2015, 2019), enables high-frequency variability to be determined by observationally-derived data, while longer-term climate change is determined by the anomaly data.

For our experimental runs (Figure 1), we used two possible future climate scenarios: The SSP1-2.6 scenario (hereafter 2C) represents a world in which the goals of the 2015 Paris Agreement are broadly achieved, leading to ~2°C warming by 2100; and the SSP3-7.0 scenario (hereafter 4C) represents a high-emission world shaped by regionalised concerns and weak global cooperation, resulting in ~4C warming by 2100 (Gidden et al., 2019; Riahi et al., 2017).

To examine the effects of forestation, we first applied the 'Max Forest' scenario developed by Roe (2021). This scenario increases forest cover by 750 Mha by the end of the 21st century (Weber et al., 2024) by expanding fractional coverage of pre-existing tree PFTs into neighbouring gridcells following climate and land-use constraints (Figure S6). Forest increases are particularly high at the margins of tropical rainforests and in temperate regions (King et al., 2024). It both represents an approximate biophysical maximum given PFT constraints (Roe 2021) and aligns with existing international initiatives (e.g. the Bonn Challenge) while protecting croplands, urban areas, and biodiversity-rich areas (King et al., 2024, Weber et al 2024). Simulations with Max Forest were performed under both 2C and 4C climate trajectories to assess how extensive forest expansion interacts with climate change ('2C Max Forest', '4C Max Forest').

For comparison, we performed two additional land use experiments reflecting plausible socio-economic futures (SSP1 and SSP3), each combined with their respective climate pathway (2C and 4C) to produce reference runs. The SSP1 results in an end-

of-century increase in forest cover of ~310 Mha, whereas SSP3 results in a decrease of ~290 Mha over the same period (Weber et al., 2024) (hereafter '2C Forestation' and '4C Deforestation', respectively). Finally, to isolate the effects of climate from those of land use change, we included control experiments in which LULC was fixed at 2026 levels from SSP2, while climate evolved following the 2C and 4C scenarios ('2C No LULCC' and '4C No LULCC').

All primary simulations were performed with population density fixed at 2026 levels from SSP2. To isolate the contribution of population dynamics, we designed three experiments under 2C warming which all followed the Max Forest LULC scenario. In the control experiment ('2C MF NoPop'), population density was also fixed at 2026 values from SSP2, while in the other two comparison scenarios, population density evolved following SSP1 ('2C MF SSP1Pop') and SSP3 ('2C MF SSP3Pop') respectively. Population projections are derived by linear interpolation from historical data (Goldewijk et al., 2017). SSP1 represents a moderate global population growth scenario and SSP3 a strong global population growth scenario dominated by increasing population in sub-Saharan Africa with population declines in parts of the global North. Similarly, to isolate the effect of fire on the global terrestrial carbon sink, we repeated selected experiments with the fire module switched off (section 3.4).

Figure 1 – Experimental procedure used in this study, showing model spinup and details of each experiment.

## 2.3 Fire model evaluation

As a participant in FireMIP, CLM5 has undergone extensive evaluations against observationally-derived fire datasets (Li et al., 2019; Hantson et al., 2020) as well as

other models using standardised protocols (Rabin et al., 2017). In the FireMIP ensemble, CLM5 was one of the best performing models for simulating spatial patterns of burned area and fire emissions, and showed good skill in capturing the seasonality of fire activity at regional scales (Hantson et al., 2020).

Since our experimental setup was performed at a higher spatial resolution and slightly different historical period than the FireMIP baseline (Hantson et al., 2020), and on a different HPC platform with different processor architecture (which can affect results under certain conditions; Guarino et al., 2020), we performed our own evaluation. We compared burned area and total fire C emissions from our historical and SSP2-4.5 simulations (2000-2016) against two versions of the Global Fire Emissions Database (GFED); GFED4.1s (Van Der Werf et al., 2017) and GFED5 (Chen et al., 2023), which are based on satellite observations.

CLM5 reproduces reasonably well the global burned area patterns observed in both GFED products, with dominant signals located in African and South American tropical grassland regions (Figure 2A). However, CLM5 underestimates burning in northern Australia and Southeast Asia (the latter primarily in GFED5). Globally, burned area in CLM5 is similar to GFED4.1s (1.3% smaller), but is substantially lower (by 41.6%) than GFED5, which accounts for small fires and uses updated satellite data (Chen et al.,

Figure 2 - Annual means of burned area (A) and total carbon fire emissions (B) in CLM5, GFED4.1s, and GFED5, over the period 2000-2016. Dotted green lines indicate tropical and boreal latitude bands used for further model evaluation (Figures S1-S4).

- 2023). These small fires are not well reproduced at CLM5's 1° resolution. A similar pattern is seen for fire emissions (Figure 2B), where CLM5 is able to reproduce the spatial distribution of fire carbon emissions, capturing Amazonian fires, but underestimating emissions from African savannas. Globally, CLM5 estimates are close to GFED4s but about 30% lower than GFED5.
- For trends (Figures 3A and 3C), CLM5 performs well compared to GFED4.1s, with the simulated burned area falling within the standard error of the GFED4.1s mean. Compared to GFED5, however, CLM5 underpredicts burned area (Figure 3A) and shows a weaker declining trend. The recent declining trend in global burned area is more evident in GFED5 than the other datasets. For fire emissions, CLM5 again performs well compared to GFED4.1s until 2016, including a peak in 2010 that does not appear in GFED5. Post-2015, CLM5 fire emissions rise, bringing them closer to GFED5 values, although the satellite-derived GFED emissions do not show a comparable increase in this period (Figure 3C). This model shift may reflect the transition from historical to SSP2 LULC after 2015.
- In the seasonal cycle of burned area (Figure 3B), both GFED products simulate peaks in boreal winter (December-January) and summer (July-September). While the monthly magnitude of global burned area in CLM5 is again comparable to GFED4.1s

Figure 3 - Trends in global burned area (A) and fire emissions (C), and seasonal cycles of the same variables (B, D) from CLM5, GFED4.1s, and GFED5. Seasonal cycles were computed over the period 2000-2016 owing to data availability for GFED4.1s. Shading in A and C represents the standard error of the annual mean for each year. Shading in B and D represents the standard error of the monthly mean across the 2000-2016 time period.

and lower than GFED5, the seasonal cycle has a delayed peak in October rather than August, and does not show the December-January peak; rather, a secondary peak is seen in boreal spring. A similar delayed peak is seen in fire emissions (Figure 3D), where the CLM5 peak is about two months later than the GFED products. CLM5 does simulate a secondary increase in fire in boreal spring which is seen in GFED5, but is again unable to capture increases in burned area in boreal winter. This discrepancy was also observed by Bhattarai et al. (2025).

Overall, CLM5 is better able to capture the global totals and broad spatial distribution of fires than their seasonality, but it does have important strengths. For example, CLM5 is the only FireMIP model able to reproduce the bimodal seasonality found in observationally-derived fire products due to its inclusion of cropland fires (Hantson et al., 2020). We also note that there is often disagreement between observationally-derived global fire datasets (Parente et al., 2016; Khairoun et al., 2024) due to varying methodologies and retrieval complexities (Hantson et al., 2013), which represents a challenge for the evaluation of fire models (Hantson et al., 2016). Evaluation of the model's performance on regional scales is available in the Supplementary Information (Figures S1-S4).

#### 3. Results

# 3.1 Global changes in burned area and fire emissions under forest expansion

In the absence of land use change, burned area decreases between 2015-2050 followed by an increase between 2050-2095, with a small overall increase of 0.01 million km<sup>-2</sup> yr<sup>-1</sup> under 2C No LULCC and a much stronger increase of 0.29 million km<sup>-2</sup>

<sup>2</sup> yr<sup>-1</sup> (+6.5%) under 4C No LULCC by 2095 (Figure 4). This represents the largest increase of any scenario.

Figure 4- changes in burned area (A, C) and fire carbon emissions (B, D) under different climate and land use scenarios. Panels A and B show global decadal means at 2015 (2010-2020), 2050 (2045-2055), and 2095 (2090-2100), with error bars representing the standard error of the decadal mean. Values are offset by ±1 year to more clearly show the error bars. Panels in C and D show spatial differences between the 4C Max Forest and 4C No LULCC experiments at 2095. Stippling indicates statistically significant differences at 95% confidence using a two-sided Student's T test. Zonal mean in panels C and D show results for 2C (blue) and 4C (red) scenarios

Both 2C and 4C Max Forest scenarios show declines in burned area between 2015 and 2050, followed by increases towards the end of the century. 2C Max Forest ends the century with a net decrease in burned area (-0.28 million km<sup>-2</sup> yr<sup>-1</sup>; -6.3%), representing the lowest burned area of any scenario; 4C Max Forest results in a moderate increase (+0.20 million km<sup>-2</sup> yr<sup>-1</sup>; +4.5%) relative to 2015 levels. The 2C Forestation scenario (i.e. SSP1-2.6) has a similar net increase in burned area as 4C Max Forest (+0.18 million km<sup>-2</sup> yr<sup>-1</sup>; +4.0%), while the 4C Deforestation (i.e. SSP3-7.0) scenario has a slight net decrease (-0.04 million km<sup>-2</sup> yr<sup>-1</sup>; 0.9%). The mid-century decreases in the Max Forest scenarios, and the 2095 for 2C Max Forest, are statistically significant, that is, values are greater than 1 standard error from the 2015 mean.

Fire emissions follow a broadly similar pattern. Most scenarios show decreases between 2015 and 2050, followed by increases from 2050 to 2095. The exception is 4C Deforestation, where emissions peak in 2050 and remain the highest through the end of the century. By 2095, all scenarios show lower fire emissions than in 2015. 2C Max Forest has the largest absolute reduction (-0.58 PgC yr<sup>-1</sup>; -24.3%). Uncertainty in

https://doi.org/10.5194/egusphere-2025-5267 Preprint. Discussion started: 12 November 2025 © Author(s) 2025. CC BY 4.0 License.

fire emissions is generally smaller than for burned area, resulting in more statistically significant changes over time.

To assess how large-scale forestation affects fire activity under identical climate conditions, we compare 4C Max Forest and 4C No LULCC in Figures 4C and 4D. Significant decreases in burned area and fire emissions occur in the margins of tropical forests, particularly in tropical Africa, Brazil, and northern Australia, in 2095. These locations correspond to where forests expand into grassland biomes in the Max Forest scenario (King et al., 2024) and dominate the global burned area and fire emissions signals (Figure 2).

By contrast, statistically significant increases in burned area and fire emissions are found in southern Africa, the Sahel and the Mediterranean, coastal South America, and parts of central and east Asia and central North America. These regions are more temperate or transitional in climate and show increased fire sensitivity to forestation under warming (Turco et al., 2018, 2023).

To further explore the relationship between forestation and fire activity across regions, we evaluated the spatial correlation between changes in forest cover and burned area under the 2C Max Forest scenario (2026-2100). Results for 4C Max Forest and for fire emissions (not shown) are highly similar. Significant negative correlations are found in tropical ecosystems correlations (*r* ~ -0.6; Figure 5), particularly in the Amazon, Congo, Southeast Asia, and northern Australia, indicating that the extensive forest expansion in these regions is associated with decreases in fire.

Figure 5 – Pearson correlation coefficients between changes in tree cover and burned area for the Max Forest scenario under 2C warming (2C Max Forest; 2026-2100). Stippling indicates grid cells where the correlation is statistically significant at 95% confidence level.

Positive correlations are found in temperate and subtropical zones ( $r \sim +0.6$ ), including the Mediterranean, central Asia, central and western North America, parts of west and southern Africa, coastal regions of South America, and the east coast of Australia. In these regions, forest expansion tends to increase fire activity.

Overall, the fire response to forest expansion exhibits strong latitudinal dependence. Increased tree cover is generally associated with decreases in fire activity primarily (though not exclusively) in tropical latitudes, whereas the opposite is the case in temperate latitudes. Note that these correlations only indicate areas where forest cover changes in the Max Forest scenario; forest-dominated regions that remain unchanged, such as the central Amazon, parts of Southeast Asia, and boreal forest regions in Russia, Canada, and Alaska, do not undergo forestation in this scenario (King et al., 2024).

#### 3.2 Regional responses of fire emissions to forest expansion and climate

To further investigate how forest expansion affects fire carbon emissions across diverse landscapes, we examine regional trends under different climate and land-use scenarios. We use the GFED region classification (Van Der Werf et al. 2010; Figure S5). Figure 6 shows projected changes in fire emissions through the 21<sup>st</sup> century in four GFED regions (Europe, North Africa/Middle East, the continental USA and sub-Saharan Africa) characterized by distinct forest types and climate-fire dynamics. These regions range from temperate forest (Europe, USA), semi-arid to arid with montane

# forests (North Africa/Middle East) and tropical/subtropical forests and grasslands (Sub-Saharan Africa) with differing fire responses to climate

Figure 6 - Fire carbon emissions by region under different climate and land use scenarios: Europe (A), North Africa/Middle East (B), continental USA (C) and sub-Saharan Africa (D). Decadal means are shown for 2015 (2010-2020), 2050 (2045-2055) and 2095 (2090-2100). Error bars show the standard error of the decadal mean. Values are offset by ±1 year for visual clarity.

In both Europe (Figure 6A) and the continental USA (Figure 6C), where temperate forest dominates (i.e. broadleaf deciduous temperate and needleleaf evergreen temperate tree PFTs), fire emissions follow the same trajectory. All scenarios show increases between 2015 and 2050, with little difference between 2C and 4C warming by mid-century. Under 2C warming, emission remain relatively stable from 2050 to 2095, while 4C scenarios continue to rise. Max Forest scenarios produce the highest fire emissions at both warming levels.

By 2095, 2C Max Forest emissions increase by 24.1 TgC yr¹(+70.6%) in Europe and 15.5 TgC yr¹(+34.8%) in the USA compared to 2015. In Europe, both 4C Max Forest (53.9 TgC yr¹; +158.1%) and 4C Deforestation (82.4 TgC yr¹; +140.2%) result in more than doubling of fire carbon emissions by the end of the century. In the USA, only the 4C Max Forest scenario (40 TgC yr¹; +90.3%) produces a comparable near-doubling in emissions.

In North Africa and the Middle East (Figure 6B), forest expansion is strongly associated with increasing fire emissions. Both 2C Max Forest and 4C Max Forest produce

https://doi.org/10.5194/egusphere-2025-5267 Preprint. Discussion started: 12 November 2025 © Author(s) 2025. CC BY 4.0 License.

significantly higher fire emissions than the other land-use scenarios under the same warming levels by 2095. The 2C Max Forest emissions in this region are comparable to 4C Deforestation. By 2095, 2C Max Forest emissions are 21.1 TgC yr<sup>-1</sup> (+89.8%), while 4C Max Forest shows an increase of 48.1 TgC yr<sup>-1</sup> (+204.2%), highlighting the strong fire sensitivity of this region to forestation.

A contrasting pattern is sub-Saharan Africa (Figure 6D). This domain combines Northern Hemisphere Africa and Southern Hemisphere Africa GFED regions and is dominated by tropical forest (i.e. broadleaf evergreen tropical tree PFTs). In this region, only 4C Deforestation shows an increase in fire emissions between 2015-2050 (354.21 TgC yr<sup>-1</sup>; +49%) associated with widespread deforestation in the mid-21st century. However, emissions then decline toward 2095. In contrast, both Max Forest scenarios show the largest declines between 2015 and 2095: 2C Max Forest decreases by 222.91 TgC yr<sup>-1</sup>(-30.8%) and 4C Max Forest by 111.51 TgC yr<sup>-1</sup>(-15.4%), indicating the negative response of wildfire to tropical forest expansion across climate scenarios.

#### 3.3 Human influence on future fire activity

To evaluate how human drivers of fire may affect fire activity under forest expansion, we focus on the Max Forest LULCC scenario under 2C warming, comparing three population trajectories: a control with fixed population (NoPop), a dynamic scenario reflecting sustainability (SSP1Pop) and a high growth scenario with intense population pressure (SSP3Pop) (section 2.2).

Figure 7– Effect of population density change on global burned area (A, C) and fire carbon emissions (B, D) under 2C warming and Max Forest LULCC. Decadal means are shown for 2015 (2010-2020), 2050 (2045-2055) and 2095 (2090-2100). Error bars show the standard error of the decadal mean. Plots C and D show spatial differences in 2095 between 2C Max Forest scenarios with changing population growth (SSP1Pop and SSP3Pop) and a control scenario in which population is fixed at 2026 SSP2 values (NoPop). Stippling indicates significant changes at the 95% confidence level (2-sided Student T test). All values are decadal means.

In the NoPop scenario, the combined effect of climate warming (2C) and forest expansion results in a modest increase in global burned area of 0.08 million km<sup>-2</sup> yr<sup>-1</sup> (+1.8%) by 2095 (Figure 7A). SSP1Pop shows an initial decline in burned area between 2015-2050, followed by recovery by 2100, with end of century totals comparable to 2015. By contrast, the SSP3Pop scenario drives significant decreases with global burned area in 2095 0.67 km<sup>-2</sup> yr<sup>-1</sup> (-15.3%) lower than in 2015.

Fire emissions (Figure 7B) follow a different trajectory, with a sharp decrease to 2050, followed by a small recovery by 2095. Differences between SSP1Pop and NoPop are minimal, while SSP3Pop results in lower fire emissions than SSP1Pop and NoPop (-0.52 TgC yr<sup>-1</sup>; -23.4%). The differences in response between burned area and fire emissions may be due to the more direct effect of population on burned area in the model; there is also a greater impact of population growth in tropical and subtropical regions, which dominate the burned area signal (Figures 7C and 7D).

Spatially, population changes produce strong regional variations in fire activity (Figures 7C and 7D). In SSP1Pop, population changes lead to significant localized increases in burned area in Central America and northern and southern China, with additional increases in fire emissions across southeast Asia, India, western South America, eastern Europe, and Zimbabwe because of population decreases. However, both burned area and fire emissions decrease in the Sahel region and East Africa, where population increases.

In SSP3Pop, widespread fire reductions occur across tropical Africa, as well as in Mexico, Afghanistan, and parts of India, which dominate the global signal. However, statistically significant increases in fire activity are found in parts of southern Europe and China, associated with a declining population which limits fire suppression capacity.

We explore regional variations in fire responses to population change in Figure 8, focusing on 4 key regions in which the same population scenarios resulted in differing fire responses: Europe, Central America, Sub-Saharan Africa, and North Africa/Middle East.

Figure 8 – Regional burned area responses to population change under 2C of warming and Max Forest LULCC in Europe (A), Central America (B), sub-Saharan Africa (C) and North Africa/Middle East (D) based on GFED regional definitions (Van Der Werf et al., 2010). Error bars show the standard error of the decadal mean.

In Europe (Figure 8A), burned area increases under all population scenarios until midcentury, after which the trends diverge. Under NoPop, burned area increases by 0.03 https://doi.org/10.5194/egusphere-2025-5267 Preprint. Discussion started: 12 November 2025 © Author(s) 2025. CC BY 4.0 License.

million km<sup>-2</sup> yr<sup>-1</sup> (+42.9%) between 2050 and 2095. The SSP1Pop and SSP3Pop scenarios show stronger increases by 2095, with SSP3Pop resulting in twice as high (0.07 million km<sup>-2</sup> yr<sup>-1</sup>; 100%). These increases are associated with projected decreases in population density throughout Europe (-4% in SSP1, -29% in SSP3; Lawrence et al., 2022), potentially reducing fire suppression capacity and contributing to higher fire activity.

In Central America (Figure 8B), SSP1Pop results in a burned area increase of 0.03 million km<sup>-2</sup> yr<sup>-1</sup> (+20.0%) relative to 2015, whereas SSP3Pop yields a decrease of 0.01 million km<sup>-2</sup> yr<sup>-1</sup> (-6.7%). This reflects opposing population projections, with SSP1 a 20% decline, and SSP3 a 77% increase during the 21st century (Lawrence et al., 2022).

In Sub-Saharan Africa (Figure 8C), burned area significantly decreases regardless of population in response to forest expansion, with the strongest reductions in the SSP scenarios. By 2095, burned area decreases by 0.56 million km<sup>-2</sup> yr<sup>-1</sup> (-26.7%) under SSP1Pop and by 0.94 million km<sup>-2</sup> yr<sup>-1</sup> (-44.8%) under SSP3Pop. These reductions align with projected population growth in the region, which is expected to increase fire suppression: under SSP3, population density rises from 37.2 people km<sup>-2</sup> in 2014 to 128.4 people km<sup>-2</sup> in 2100 (+245%), while SSP1 projects a smaller but still substantial increase to 65.8 people km<sup>-2</sup> (Lawrence et al., 2022). In contrast, North Africa/Middle East (Figure 8D) show an increase in burned area across all scenarios, regardless of population projection. The strongest increase occurs under SSP1Pop, where burned area doubles between 2015 and 2095, and remains significantly higher in 2095 than in NoPop and SSP3Pop scenarios.

# 3.4 Carbon sequestration impacts of fire emissions under forest expansion

The global significance of fire as a key component of the terrestrial biosphere means that any estimates of land-based CDR potential should take its impacts on carbon storage into account. To evaluate this, we compare our model experiments using the CLM5 fire module with identical experiments in which the fire module is switched off.

Figure 9 shows that fire significantly reduces the terrestrial C sink (comprising vegetation, soil, and litter carbon) by the end of the 21st century under both warming scenarios (2C and 4C). Reductions in land C are widespread across all continents, with impacts greatest in tropical Africa. Other regions with important reductions include Brazil, Argentina, and Venezuela; Mexico and the western United States; the Mediterranean basin (Portugal, Spain, Greece, and Turkey); Russia, Mongolia, and

Figure 9– Effect of fire on terrestrial biosphere carbon. A – difference in land C between 4C MF and 4C MF without fire, averaged over 2090-2100. Stippling indicates statistically significant differences at 95% confidence. The zonal mean plot shows results for 4C (red) and 2C (blue). B – timeseries of the evolution of land C for paired fire/no fire experiments across all scenarios, with a 20-year running mean applied for smoothing. Shading indicates the difference due to fire.

To understand how fire shapes the trajectory of carbon accumulation over time, we compare experiments with and without fire across different climates (2C and 4C) and land use scenarios (Max Forest and No LULCC) (Figure 9B). The greatest carbon gains occur in scenarios combining high CO<sub>2</sub> and extensive forestation, yet these gains are reduced when fire in included. For example, in the 4C Max Forest scenario, the land carbon sink reaches 516 PgC (+17.6% compared to 2015) in the absence of fires, but only 452 PgC (+15.7%) when fire is included, which is a net reduction of 64 PgC by the end of the 21<sup>st</sup> century. Similar reductions (~60 PgC) are observed across all experiments, irrespective of climate and LULC scenario. To place this in context, the estimated average annual anthropogenic CO<sub>2</sub> emissions over the period 2014-2023 is

https://doi.org/10.5194/egusphere-2025-5267 Preprint. Discussion started: 12 November 2025 © Author(s) 2025. CC BY 4.0 License.

10.8 PgC yr $^{-1}$  (Friedlingstein et al., 2025). Therefore, by the end of the 21st century, the cumulative impact of fires on the land C sink is equivalent to ~5.6 years of global anthropogenic CO $_2$  emissions.

To examine regional differences in how fire affects terrestrial carbon storage, we
disaggregate the land carbon sink at the middle (2045-2055) and the end of the 21<sup>st</sup>
century (2090-2100) in Figure 10, following the GFED regional classification (Figure
SX). The greatest fire-induced reductions in land carbon sink by 2100 occur in
southern hemisphere Africa (SHAF), boreal Asia (BOAS) and central Asia (CEAS). In
SHAF, land carbon declines by 12.0 PgC under 4C warming. BOAS has a reduction
of 11.5 PgC under 2C, while CEAS shows a reduction of 11.8 PgC under 4C. Each of
these losses is greater than current annual anthropogenic CO<sub>2</sub> emissions
(~10.8 PgC yr<sup>-1</sup>; Friedlingstein et al., 2025). Other regions with large reductions

include northern hemisphere Africa (NHAF; 6.8 PgC under 4C) and southern hemisphere South America (SHSA; 6.9 PgC under 2C).

Figure 10– Regional differences in the impact of fire on the land C sink (vegetation, soil, and litter C) at mid-century (2045-2055; left) and end-of-century (2090-2100; right), disaggregated by GFED regions (Van Der Werf et al., 2010): the Americas (A and B), Europe, the Middle East, and Africa (C and D), and Asia and Oceania (E and F). Bars represent the difference in total land C between simulations with and without fire. Blue bars correspond to the 2C MF scenarios; red bars correspond to the 4C MF scenarios. Error bars represent the standard error of the decadal means. Region definitions are as shown in Figure S5: BONA = Boreal North America, TENA = Temperate North America, CEAM = Central America, NHSA = Northern Hemisphere South America, SHSA = Southern Hemisphere South America, EURO = Europe, MIDE = North Africa/Middle East, NHAF = Northern Hemisphere Africa, BOAS = Boreal Asia, CEAS = Central Asia, SEAS = Southeast Asia, EQAS = Equatorial Asia, AUST = Australia/New Zealand.

#### 4. <u>Discussion</u>

This study quantifies how large-scale forest expansion, climate change and population growth interact to affect global fire activity and the terrestrial carbon sink. Our experiments show that forest expansion impacts burned area and fire carbon emissions in regionally varying ways, independent of the climate warming scenario.

In tropical latitudes, especially around the margins of tropical rainforests, forest expansion is associated with decreasing fire activity (Figures 4 and 5). In the Max Forest scenario, tropical forest expansion comes at the expense of C4 grasslands, such as those in the Brazilian Cerrado and southern African savannas, which are among the world's most fire-prone ecosystems (Bond et al., 2019). These reductions in grassland area directly suppress burned area because forests burn at a lower frequency than grasslands; the tropical tree PFTs have lower flammability than the grasses they are displacing (Li et al., 2018). A sharp decline in fire activity early in the simulation is also observed (Figure 6), especially in the tropics, which reflects the removal of deforestation fires, previously a dominant fire source. Since No LULCC and Max Forest scenarios eliminate future deforestation, these fire types are reduced almost to zero. SSP1, though generally a forest expansion scenario, does still include some deforestation (Loughran et al., 2023). The implication of this finding is that halting and reversing deforestation, which was a key commitment made by parties to the United Nations Framework Convention on Climate Change at COP26 in 2021 (Wang et al., 2022), could substantially decrease tropical fire activity.

However, forest expansion can also increase fire. In temperate regions such as the Mediterranean and central Asia, forest expansion leads to increased fire activity (Figure 6), especially under high warming. These regions are projected to became drier and hotter, increasing fuel flammability. The Mediterranean region is of particular concern due to its dense population, existing fire vulnerabilities and the high value of its ecosystem services. Our results suggest that forest expansion as a mitigation strategy could constitute maladaptation, as it may intensify wildfire impacts, damage infrastructure, degrade air quality and reduce the carbon storage potential of forests (Turco et al., 2018).

Our results also highlight the importance of population change as a key driver of future fire activity (Zhang et al., 2025; Veira et al., 2016). On a global scale, the magnitude of the fire response to population scenarios is similar to that from warming and LULC change (Figure 7). In sub-Saharan Africa, population growth under SSP3 results in a strong reduction in burned area, primarily through increased fire suppression associated with rising population density (Figure 8). A similar but smaller suppression signal occurs in SSP1. These scenarios contrast with Europe and East Asia, where population decline is associated with increases in fire activity, as has been observed in recent decades (de Diego et al., 2023). Our model assumes that fire suppression scales with population density, capturing the human influence on fire ignition and control (Li et al., 2012). Regional examples such as central America and North

Africa/Middle East highlight how divergent population trajectories under SSP1 and SSP3 drive opposing fire trends (Figure 8).

Our analysis shows that fire activity is driven by complex interactions between land use, climate, and population, and not by warming alone. Across several regions, fire-related outcomes under 2C and 4C warming scenarios were of similar magnitude (eg, Figures 6 and 10). Fuel biomass and soil moisture have been shown to be important determinants of fire behaviour that are not functions of temperature (Turco et al., 2018), but we found that the increased biomass and decreased soil moisture resulting from tropical afforestation in the Max Forest scenario (King et al., 2024) did not increase fires locally because the signal was instead dominated by halting and reversing deforestation. Bhattarai et al. (2025) found decreases in tropical fire activity with constant land use under 2C and 4C warming scenarios, with stronger decreases under 4C, attributing this to complex non-linear relationships between fire and precipitation. We note that projections of future tropical precipitation, particularly in Africa, are highly uncertain (Taguela et al., 2025).

In addition to burned area and fire emissions, we found that fire substantially reduces the carbon sink across all scenarios by the end of the 21<sup>st</sup> century (Figure 9). This has implications for climate mitigation planning. Our results show failing to account for future fire activity can lead to overestimate of carbon sequestration potential. For example, in boreal Asia, where some countries, such as Russia, rely heavily on forest carbon sequestration in their NDCs (Kurichev et al., 2023), omitting the impact of fire could result in an overestimation of land C sink by more than 10 PgC by 2100 (Figure 10). More broadly, this suggests that NDCs under the Paris Agreement must consider future fire dynamics to realistically assess land-based mitigation contributions (Burton et al., 2024).

#### Model caveats and uncertainties

Several caveats apply to this study. First, we used a single model. While this fire framework has been extensively used and evaluated (Teckentrup et al., 2019; Hantson et al., 2020; Lasslop et al., 2020), it represents only one approach to the complex task of mechanistic fire modelling. Our results therefore may partly reflect model-specific responses. Model intercomparison studies, such as FireMIP, are thus vital for better understanding the range of potential differences in fire models and assumptions. In particular, comparing future fire projections across climate and LULC scenarios can help constrain uncertainty on the role of fire in climate mitigation and adaptation.

Second, our simulations were forced with climate forcings from a single Earth system model, CESM2. Sensitivity to the climate forcings could be tested by driving the fire model with a range of climate forcings derived from ESMs with varying climate sensitivity. Future work could evaluate fire responses by forcing fire models with post-

https://doi.org/10.5194/egusphere-2025-5267 Preprint. Discussion started: 12 November 2025 © Author(s) 2025. CC BY 4.0 License.

processed ESM outputs, as recommended by Li et al (2024), to improve the reliability of their projections.

Third, the use of transient land use change scenarios constrained vegetation dynamics. While these scenarios are necessary to investigate the impacts of land use change over time and the fire module feedbacks to the carbon pool, they do not allow vegetation responses (e.g. post-fire regrowth, succession) to be simulated dynamically. Further work could explore these dynamics by employing a fire module within a dynamic vegetation model with prescribed LULC at different future time-slices and applying constant climate forcings.

Finally, our Max Forest scenario assumes that forest expansion occurs through expanding existing, locally dominant natural tree PFTs into surrounding grid cells (Roe, 2021). It therefore avoids introducing non-native species, and that approach may not reflect real-world situations where expansions in tree cover are achieved through commercial plantation forestry using non-native species (Lewis et al., 2019). These practices can result in adverse fire implications; for example, in Portugal, introduction of non-native *Eucalyptus* species combined with plantation abandonment results in high fire hazards in the absence of intensive management (Tomé et al., 2021), while the expansion of commercial forestry and oil palm plantations in parts of Indonesia has been associated with increased fire impacts (Marlier et al., 2015; Purnomo et al., 2018). Timber plantations in temperate regions are twice as likely to burn as native forests (Bousfield et al., 2025). Our scenario thus likely underestimates fire risk associated with plantation-style forest expansion.

#### 6. Conclusions and outlook

Our findings highlight that forest expansion interacts in complex ways with climate and population to affect both global and regional fire regimes. In tropical regions, afforestation/reforestation tend to reduce fire activity locally, both by removing the deforestation fire signal, and by replacing highly flammable grasslands with less flammable tropical tree cover. In contrast, forest expansion in temperate regions (e.g. the Mediterranean and continental US) tends to increase fire activity, due to increased fuel availability, drier conditions, and decreased fire suppression.

Population changes can exert a regionally comparable influence on future fire activity to that of climate and land use. For example, population density growth in sub-Saharan Africa enhances fire suppression and decreases burned area, whereas population declines in Europe may contribute to increasing fire activity.

Beyond its impacts on ecosystems and carbon sequestration, fire has important effects on air quality and human and vegetation health (e.g. Val Martin et al., 2015; Ford et al., 2018). Forest expansion scenarios such as those explored in this study could change fire emissions in densely populated regions, potentially improving air quality in some areas while worsening it in others. Building on this work to quantify air pollution and

health outcomes will be essential to fully assess the benefits and trade-offs of forest expansion as an effective nature-based solution under climate change.

Fire also significantly decreases the land carbon sink by the end of the 21st century, as much as ~60 PgC across all scenarios, underscoring the need to include fire dynamics in global and national climate assessments. Failing to account for future fires could lead to overestimates of land-based carbon sequestration, especially in countries relying on forest sinks in their NDCs.

#### 7. Acknowledgements

This work was funded by a United Kingdom Research and Innovation (UKRI) Future
Leaders Fellowship awarded to MVM (grant number MR/T019867/1). Model
experiments were performed on the UK national supercomputing service ARCHER2.
Data analysis was performed using JASMIN, the UK's collaborative data analysis
environment. S. Roe and P. Lawrence originally developed the 'Max Forest' scenario.
The authors thank O. Haas for helpful discussions and J. Weber for assistance
preparing the 'Max Forest' input data.

# 8. Availability Statement

Land use timeseries data used to run the experiments and model outputs used in the analysis are available on Zenodo: land use input data (DOI: 10.5281/zenodo.17424044); CLM5 output data (DOI: 10.5281/zenodo.17424426).

#### 9. Author contributions

JAK and MVM conceived the study. JAK designed and performed model experiments, analysed the outputs, and wrote the original draft. MVM contributed to the development of the manuscript and provided project funding and administration. KJS and CGB contributed to the development of the manuscript.

# 10. Competing interests

JAK is on an advisory panel for Ecologi, an organisation which invests in ecosystem restoration projects.

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
