# Peer review of "Global and Regional Impacts of Forest Expansion on Future Wildfires"

_EGUsphere, 2025_

## Referee Comment (RC1)

**General Assessment:**

The research paper *Global and Regional Impacts of Forest Expansion on Future Wildfires* provides analysis of land simulations that include mechanistic fire modelling to quantify the balance and interaction of effects of climate change, large scale afforestation and reforestation, and population changes on fire.

The simulation setup is convincing and comprehensive for the research questions posed. I appreciate the value and effort to start with a meaningful model evaluation. The methods of the analysis beyond this point however seem preliminary and urgently warrant improvement. The presentation in figures and text seems raw, unpolished and hard to understand, it is often overdetailed on numeric results and superficial in interpretation. Overall, it is an unsatisfying use of the nevertheless useful data for the answer to the research questions. However, seeing the potential of the data and the analysis to grow to a meaningful and useful level that expands our knowledge of land use–climate interactions, I suggest the authors thorough changes to their analysis and presentation.

**Major Comments:**

**The term *nature-based solution* (e.g. line 38, 86)**

This is a widely applied term, but in the context of climate change mitigation, it is especially misleading. Indeed, some scholars frame forestation and similar land use changes as "solutions", but literature clearly shows that these cannot replace emission-reduction to reduce net emissions fast enough to keep limits defined in the Paris Agreement in reach. Especially given your results, a more critical framing, e.g. with "solution" in quotation marks is justified to avoid the reproduction of such misleading framings.

**Forest expansion halts deforestation (e.g. line 46/47, 530-538, 567-569, 625/626)?**

It is hard to grasp what you mean. In most literature and in practice, avoided deforestation is discussed as distinct from forestation. With this as background, please reformulate or avoid entirely to clarify. Otherwise, what you present as finding seems to be trivial, as it is basically in the scenario assumptions. If there is an important reason to keep it, please rephrase it to transparently show the triviality or what else you deem to stress here.

**Misleading comparison of cumulative carbon sink reduction to present-day emissions**

In understand the desire to communicate the carbon flux in terms of well-understood present-day emissions. However, please use great care with comparisons of temporally integrated carbon uptake potential with emission rates. Such claims of "equivalency" artificially suggest that 5.6 years of current emission are added from the land carbon budget until 2100. It is precisely the comparison of integrated flux with flux rate that misleads here: The carbon released by fires in expanding forests until 2100 enter the atmosphere at different times in the chronic of global warming then present-day emissions and has different impacts on peak warming, overshoot duration, and so on. In awareness of the sadly widespread usage of this misleading comparison, I suggest the following: In the abstract, use the maximum over the period 2015-2100 of annual emission rate of additional fire. Complementarily, in the discussion, you can draw this comparison and debunk the misleading typical claim of equivalency of cumulative carbon uptake over the 21st century with present-day emissions. Moreover, a comparison with other cumulative carbon sink changes will be informative (see below).

**Deeper and more ambitious analysis and interpretation needed**

For large parts of the paper, the authors have accumulated statements about findings that are not novel at all. For a large part, they are directly deducible from the model structure, without simulations.

For example, this is the case for like "Our results also highlight the importance of population change as a key driver of future fire activity" (line 548) and relate it to "Our model assumes that fire suppression scales with population" (line 556/557). It remains unclear however, where the analysis of the simulation output actually provides new learnings beyond the assumptions introduced to the study in the model structure.

However, this is also the case for the effect of fire on long term mean carbon stocks (cumulative sinks). The authors only state the obvious, the sign ("reduction"), and quantify the reduction in stock (roughly 60 PgC) for one scenario, which is expected to be a particularly model-dependent metric. However, they neither provide information on systematics in scenario-dependence of this value, nor do they adequately contextualize it as fraction of changes in stock from climate change, land use change or population change, nor do they quantify the size of interaction terms.

Deeper and more ambitious analysis is needed to use the simulations for added learning, which they are clearly potent for, in particular the quantitative assessment of relative contributions of population, forest cover and climate changes to fire changes in maps and aggregated in meaningful regions. Below, where applicable, I tried to suggest specific additional analysis and steps one could undertake to enhance the value of the presented analysis.

**Minor and linguistic comments:**

Line 49/50: this is an interesting finding. For a stronger statement, please consider reporting the factor applying in low-warming scenarios. If even in such moderately heated world, fire emissions increase drastically, readers will be able to anticipate the even more compromising effects in a strongly heated world.

Line 53: "Finally, fires reduce, …."

- "Overall" instead of "Finally"?
- the effect of fire (not to be confused with the fire in added forests)
-
Line 54:

Please clarify with more explicit wording whether this is avoided carbon flux rate integrated over time (which start date?). If you aim to compare it to present-day emission rates, for clarity and interpretability in terms of negative emission potential please chose the same unit (gC/year).

Line 57: "interactions" instad of "feedbacks"?
Line 58: "climate change" instead of "CO2"? CO2 is not mitigated, climate change ideally is.

Line 62: "process in" instead of "component of".
Line 62/63 "It is one of the most significant controls on the carbon cycle": this is a bit of an overstatement. Its acting only on land, and only in certain landscapes. "one of the most significant controls of the land carbon cycles" might me more adequate.

Line 64: "feedback responses": this term is unclear to me. Do you mean the alteration of the feedback process fire?

Line 86: I propose "as a so called *nature-based solution*" This place could be the occasion to introduce the criticality of the term that wrongly presumes effectiveness.

Line 100-102:  This concern has been raised in previous literature, e.g.

Hermoso, V., Regos, A., Morán-Ordóñez, A., Duane, A., and Brotons, L.: Tree planting: A double-edged sword to fight climate change in an era of megafires, Glob. Change Biol., 27, 30013003, https://doi.org/10.1111/gcb.15625, 2021.

there has been a study without process representation pointing against the concern:

Golub, A., Sohngen, B., Cai, Y., Kim, J., and Hertel, T.: Costs of forest carbon sequestration in the presence of climate change impacts, Environ. Res. Lett., 17, 104011, https://doi.org/10.1088/1748-9326/ac8ec5, 2022.

and there has been a study raising the concern specifically for large-scale forestation-reliant mitigation pathways with forest expansion patterns similar to the one you use:

Jäger, F., Schwaab, J., Quilcaille, Y., Windisch, M., Doelman, J., Frank, S., Gusti, M., Havlik, P., Humpenöder, F., Lessa Derci Augustynczik, A., Müller, C., Narayan, K. B., Padrón, R. S., Popp, A., van Vuuren, D., Wögerer, M., and Seneviratne, S. I.: Fire weather compromises forestation-reliant climate mitigation pathways, Earth Syst. Dynam., 15, 1055–1071, https://doi.org/10.5194/esd-15-1055-2024, 2024.

Please use such literature to introduce existing knowledge on the concern that is specific about forest expansion.

Line 117/118: Jaeger et al, 2024 raised and showed this.

Line 147-150: In this context, please clarify that with this setup, one cannot consider atmospheric feedbacks to LULCC. As some of them can be important mitigators of fire (local BGP cooling in the Tropics, global BGC cooling) the reader needs to be informed about these key limitations here and in the discussion.

Line 165-167:

Maybe add one sentence about how close this brings the 2015-25 evolution to observed environmental change.

Line 177/178: this is wrong. A central goal of the PA is to limit global warming to "well below 2°C [...]". With most likely warming lying around 2°C, this is not achieved.

Line 180: add "°C" in appropriate distance from "4".

Line 183: I recommend to be precise here about a claim about the model world: "increases prescribed forest cover in the model"

Line 187 "it": Reference unclear. The 'Max Forest' scenario?

Line 191/192: please clarify in your wording that only atmospheric impacts on land, not vice versa can and are assessed.

Line 195/196: please add the reference for the land use change patterns. Is it Popp et al. (2017)?

Line 209: This is hard to understand. If it's a projection, shouln't it be extrapolation?

Figure 2 and from here on everywhere: The label says km^-2, but you probably meant km^2. But please express as grid cell areal share to avoid dependence on grid cell area and to allow for comparison between latitudes.

Line 244: from this, can we deduct the inability of CLM5 to represent the effect of small-scale fires?

Line 253/254: Do I as a reader get more information about this effect and the reasons behind somewhere else? Without more, it's unclear to which degree this is speculation or is grounded in understanding.

Line 255 "simulate": to avoid confusion with modeling data from freely running simulations, maybe use "report".

Line 263/264: please provide a statement whether this discrepancy is well explained or whether it is an obscure standing issue.

Line 283: Reference to figure: which panels?

Figure 4: Overall, to not confuse more than report, this figure needs substantial improvement.

> Line 286/287: The abrupt changes, especially from 2015 to 2050, suggest there are important evolutions inbetween. Given the fact that you simulated all the time steps, please provide the information in form of a continuous line plot. Given the close-to-linear-with-amplitude behavior of uncertainty of the decadal mean, it could be represented with shading in a year-to-year line plot (with e.g. running 10 year mean as filter) for e.g. 2 out of 6 scenarios. For interpretability, please additionally consider splitting the information in two panels, with one providing the baseline evolution (No LULCC) and another one providing the anomaly in the SSP1 and MF forestation scenarios.
>
> Furthermore, please consider using relative change as metric that allows to compare A,C with B,D appropriately.
>
> Please improve also the visual coding ogf the line - label correspondance. The markers are hardly distinguishable, in continous line plots, consider combining color and dashing/dotting for clearer distinction.
>
> Line 290/291:
>
> The stippling is
>
> - hardly visible in size
> - indistinguishable from grid lines
> - unintuitively dominant in regions of comparably small change
>
> Please improve:

- the resolution of the figure (consider vector graphic)
- the illustration of significance

Line 293-295: how does it evolve in between?

Line 295-309: please provide the uncertainty you anyway calculate with it.

Line 325 "spatial correlation": As far as I understand Fig. 5 you computed correlation across time for every grid cell. If this is true, write "spatial patterns of temporal correlation".

Line 327 "highly similar": behave similarly.

Line 328: "correlations" is doubled.

Figure 5: Consider a correction for the false discovery rate with multiple testing at this level. E.g. Hochberg-Benjamini-correction (same for all maps).

Line 336/337: Please try to avoid jargon and stick with model-explicit wordings, as what you report is model behavior, particularly modeling with one-way effects of atmos on land, without feedbacks.

Line 341/342: consider computing this metric also for SSP1 and SSP3 LULCC. Then, you could provide also estimates for other regions than the ones you alter with MF.

Line 342-345: shorten and reformulate to make this part of the sentence meaningful, otherwise it provokes the thought "of course unchanged regions do not undergo forestation" ...

Line 353: "characterized …dynamics." move up into "we use the GFED regions..."

Line 353-356: can you make this paragraph denser in information?

Figure 6: comments apply as in Fig 4. However, reading the full paper, it seems to me that no helpful information for the large findings is given here. Most important results are presented in the maps in Fig. 4 already. Consider omitting this figure entirely.

Line 366 "emission": "fire emissions" (just to be clear about driving fossil fuel emissions and responding fire emissions)

Line 366 "relatively stable": please try to be more precise in analysis (plotting see comments to Fig 4) and wording. "relatively stable" is vague for e.g. USA fire emissions under 2C, no LULCC dropping by roughly 60-80% of the 2015-2050 rise.

Line 369: Seeing the complexity of this scenario labelling, is there something structurally in your type of analysis that reduces the validity of a global warming level (GWL) analysis? This would greatly help the accessibility, brevity of summary and interpretability. As fire seems to respond quite lag-free to global / regional warming, I see no need to strictly stick with time here and strongly suggest to complement and summarize your findings in global warming space.

Line 369-374: same as above, please provide uncertainty estimates.

Line 378/379: how is this comparison relevant?

Line 383-385: is this combination helpful and meaningful? In CESM2 and CLM5 analyses of land use change impacts, often, contrasting signals could be found in West Africa vs. the tropical forest edge around the Kongo basin. This also seems to apply to your anlyses (Fig. 4 and 5). In this context, your plot depicts a residual of two strong opposing signals, which is quite hard to understand.

Line 396 "dynamic": it's not dynamic in the sense of "in interaction with climate" as far as I am concerned. Maybe choose "transient" for a clearer distinction between "changing in prescribed manner" and "responding in prognosed manner".

Line 397: is sustainability really the key aspect of the SSP1 population trajectory? As far as I understand, it's the overall label for SSP1 projections, but for population more precise terms will help to contextualize the meaning of it.

Figure 7: see Fig. 4 for comments

Line 406 "modest increase": given the calculated uncertainties shown in 7A, its insignificant. Please phrase adequately.

Line 409 "decreases with": revise for clarity

Line 411 "follow a different trajectory": this wording is confusing to me. As the processes underlying BA and fire emissions are the same, we can read from this a different relative contribution of the population signal.

Line 415: do you discuss this below? If yes, please refer me there. Otherwise: could this be due to more intense fires with carbon accumulating under human control`? Or is this represented differently in the model?

Line 418-460: please shorten this section significantly. As far as I am concerned, there are two learnings from the population experiments: 1) both fire metrics are anticorrelated with population density (probably this is quite directly implemented, hence starkly imprinted on the results) and 2) the relative population contribution to overall fire response under MF + Pop varies by region. The descriptions you provide however, are some details that for me do not seem to matter for what your paper contributes. So please consider to cut this down to key results and maybe aggregate your results like in Fig. 5 as correlation with population density.

Line 422 "However": "Consistently" would make more sense.

Figure 8: Like for Fig. 6, consider omitting this figure entirely. Otherwise, please argue for the added learning from this and improve it along the comments given for Fig. 4.

Figure 9: this figure is much nicer than the previous ones: easier to read. However, please still fix the hardly visible stippling in the map. Also, add labels and a title to the latitudinal line plot.

Line 486: please be precise. In 2015, per definition, the land carbon sink was zero. So its +17% (add uncertainty!) on carbon stock, not sink.

Line 487-489: please give an aggregated estimate of this difference with an uncertainty estimate across years / ensemble members and scenarios.

Line 489-494: as detailed for the abstract, this is inadequate context. More valuable context is SSP1,2 or 3 cumulative emissions up to 2100. Or, the most informative context is the cumulated additional carbon sink from MF. In this context, two key questions you can answer with your data, are, A) by how much is the additional land carbon sink from afforestation reduced by fire and B) how much does the importance of fire change through forestation?

So, ideally, I would like to see two values, where C stands for carbon stock:

fraction answering question A = (C_MF - C_MF_noFire) / (C_MF - C_noLULCC)

fraction answering question B = [(C_MF - C_MF_noFire)  - (C_noLULCC - C_noLULCC_noFire )] / (C_MF - C_noLULCC)

Line 500-502: see above and the general comment in the abstract. There is a manifold of more meaningful contextualizations. This one is a dangerous framing at utility to those interested in the delay of emission reduction.

Figure 10: the information in this figure is almost entirely already covered in Fig 9 and is much harder to grasp in this one. Please consider alternatively to plot above-mentioned fractions A and B regionally or--even better--as a map.

Line 522: please rephrase. What you find is actually a scenario-dependence for parts of the impacts.

Line 542-547: do you trust your results to the same degree in West Africa for example?

Line 549-551: that is not true for fire emissions, where Fig. 7 shows its clearly the smallest contributor...

Line 556-557: From this discussion I get the impression that all your results on fire response to population could be obtained without actually running the model. If this is the case, what can we actually learn additionally from these simulations? Isn't it about the balance of the drivers? I strongly suggest to discuss solely this and to avoid pretending the response of fire to population density trends is a result. It is the assumption you put in the model.

Line 567-569: this is hard to understand. Does this finding come from the comparison of MF with SSP3? In noLULCC, there should be no deforestation fire, should there? Or is this again the trivial finding that forest expansion is not deforestation?

Line 576/577: by how much? I suggest quantifying with fractions A and B as introduced above.

Line 580: please also or alternatively present a relative quantity.

Line 585 Model caveats and uncertainties:

1) Please also discuss how your findings could be affected and altered if you included the feedbacks through the atmosphere: global-scale cooling from carbon uptake under MF and spatially diverse biogeophysical feedbacks.
2) If you decide to compute fractions A and B, consider discussing how carbon prices might change based on the additional area needed to sequester a ton of carbon under fire in total (A) and fire changes from forestation (B).
3) Given this discussion section has "uncertainties" in the title, consider discussing the large global and regional mean uncertainties you quantify in the Figures. Oftentimes you report

detailed numbers when the uncertainty bars of experiment and control simulation overlap. This would be the place to summarize the meaning of uncertainties from different sources for different findings.

Line 590-594: please think one step further and summarize for the reader the consequences you expect a use of all the FireMIP models would have for your results and findings.

Line 596-598: If you follow my suggestion to analyse your results on warming levels, you can here argue that this (to a large degree) eliminates climate sensitivity as uncertainty in your analysis.

Line 619/620: well put. To argue more in favour of your approach, you could highlight that this makes your approach produce a conservative estimate that already is significant.

Line 625/626: again, this depends on the counterfactual, not on the forestation...

Line 630/631: rephrase to clarify that only in some regions the signal is of comparable size

Line 631-633: in each region, highlight the relative contribution, not the well-known gross effect.

Line 641-645: this has been argued for previously (e.g. Hermoso et al., 2021 and Jaeger et al., 2024, see comment in line 100). Cite these references and add numbers where you believe your work provides robust quantification (e.g. fractions A and B).